

# OzoNet: Atmospheric Ozone Interpolation with Deep Convolutional Neural Networks

Mohamed Akram Zaytar[1] and Chaker El Amrani[1]

[1]Department of Computer Engineering, Faculty of Sciences and Technology, Tangier, Route Ziaten, PO. Box 416, Morocco.

**Correspondence:** Mohamed Akram Zaytar (m.zaytar@uae.ac.ma)

**Abstract.** We propose a deep learning method for Atmospheric Ozone Interpolation. Our method directly learns an end-to-end mapping between classically interpolated satellite ozone images and the real ozone measurements. The model architecture represents a deep stack of convolutions (CNN) that takes the already interpolated images (Using the classical state-of-the-art interpolation method) as Input and outputs a more precise Interpolation of the Region of Interest. Our deep CNN has a lightweight structure, yet demonstrates state-of-the-art interpolation quality, and achieves optimal data processing latency ($\Delta T$) for production-ready near-real-time Atmospheric Image Interpolation, which has a big advantage over the state of the art classical interpolation algorithms. We explore different network structures and parameter settings to achieve trade-offs between performance and speed. This method showcases the potential applications of deep learning in Remote Sensing and Climate Science.

## 1 Introduction

Remote sensing data represents a big opportunity in the field of climate science and atmospheric environmental research Ustin (2004), satellites with high performing sensors provide rich and continuous streams of data about weather, agriculture, pollution, oceans among many other natural phenomena Joshi et al. (2016).

This paper focuses on Atmospheric Ozone Monitoring using remote sensing data, EUMETSAT is responsible for streaming rich data-sets in short intervals of time to the many users they have (among them us at MDEO El Amrani et al. (2012)), EUMETSAT offers ozone scans over Africa and the whole 0-degree face of the earth, a lot of this data has missing or damaged pixel ozone measurement values due to the many natural phenomena that prevent MSG's SEVIRI sensor from capturing valid ozone measurements.

On the other side, 2-D Interpolation is a classical problem in Mathematics and Computer Vision, and state-of-the-art 2-D Interpolation algorithms are well established and researched. However, research in Satellite Image Interpolation is not common, and trade-offs between Interpolation speed and Accuracy for different types of remote sensing image data might be hard to document. This paper studies the trade-offs as well as the performance of Convolution-based deep neural models and benchmarks a deep CNN LeCun et al. (1990) against the classical state of the art 2-D interpolation algorithms.





## 2 Data Pre-processing

### 2.1 Data Source

MDEO receives Remote sensing data from EUMETSAT's operated MSG satellite, and specifically from the SEVIRI instrument in MSG. EUMETCAST is responsible for broadcasting the data in fixed time intervals to all of EUMETSAT's users and
5 partners.

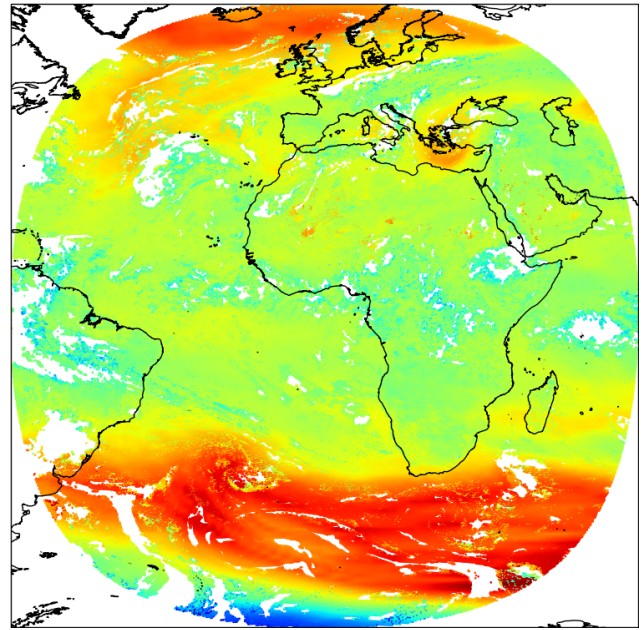

**Figure 1.** An Example of a Scan corresponding to one received file, Measurements represent the pixel ozone densities in Dobson Units.

The product responsible for the data is called "Total Ozone - MSG - 0 degree" Schmetz et al. (2002) and it has the following specifications:

- Data Type: Remote Sensing.

- Satellite: MSG (Meteosat Second Generation).

10  - Orbit type: GEO (geostationary).

- Sensor: SEVIRI (the Spinning Enhanced Visible and Infra-Red Imager).

- Sensor Type: Optical.

- Coverage: -67.5 to 67.5 degrees in Latitude and Longitude.


Every 30 minutes, MDEO receives Total Ozone compressed binary files that carry a scan over the whole 0-degree zone. After arriving, they go through a data pipeline responsible for decoding and saving new versions of the data of interest in a structured format (matrices containing the pixel level numerical measurements).

A Continuous stream of BUFR data is automatically transformed using pre-defined routines for the purposes of data analysis and visualization.

All of the experiments and results presented in this paper were based on the "Total Ozone - MSG - 0 degree" product. It represents the total density of ozone in the atmospheric column for each image segment, based on the SEVIRI 9.7 channel and other IR (infrared) and WV (Water-Vapour) channels. the SEVIRI IR 9.7 channel is sensitive to tropospheric and stratospheric in a layer between 900 hPa to 50 hPa. The Ozone column density provides information on the dynamic processes in the lower stratosphere and the upper troposphere. This product is designed to support weather forecasts, Atmospheric chemical processes research Aminou (2002).

## 2.2 Data Decoding and Preparation

MDEO receives the data in a Meteorological binary format named BUFR (the binary universal form for the representation of meteorological data), BUFR is the World Meteorological Organization (WMO) standard file format for the exchange of weather observations. It's table-driven and uses descriptors to form and send messages.

The MSG-Ozone BUFR messages provide MDEO with Ozone information on specific map pixels (or geographical points on the map) that covers the 0 degrees MSG (Latitude range: -67.5 to 67.5 degrees, Longitude range: -67.5 to 67.5 degrees), the files must be decoded and reformatted for general use.

We used a library named pybufrecmwf which provides a python interface (to ECMWF's bufrdc library) to decode the messages into Comma-separated-values files and export them into a storage unit in MDEO's servers.

The exported tabular data comes with multiple columns, we are interested in the temporal and location columns, the following columns were selected:

– Time of Measurement: Year, Month, Day, Hour, Minute, Second.

– Geographical Point: Latitude, Longitude.

– Ozone Information: Density in Dobson Units and a Quality Flag representing an indicator of the overall quality of the specific measurement.

MDEO receives one scan each half an hour corresponding to the whole MSG 0 degree zone, each scan comprise around 900,000 measurements, for Morocco MDEO receives around 25,000 measurements per scan, which covers the entirety of Moroccan land (the data is filtered so that the longitude is between -17.1 and -0.84 degrees, and the latitude is between 21.21 and 35.98 degrees).





## 2.3 Patches Extraction

The incoming data doesn't follow pre-defined rules concerning its encoded measurements, the pixels locations aren't pre-defined and the distance between them isn't constant either, and this is why a new grid needed to be constructed over the region of Morocco to normalize any incoming scan using it. Here is an overview of the steps taken to Create the final Grid:

An equidistant fine grid capable of containing all geographic points of 2 precision was created, after experimenting with different grid parameters, we settled on a grid with a step of $0.001$ degrees in latitude and longitude (equivalent to a neighborhood or a street).

Secondly, and for any incoming scan, Max-Pooling was used to project the scan over the grid, and to reduce the dimensionality of the fine grid and turn it into an almost filled image. Using a block reducer of 5x5, we were able to get full maps that

represent the Ozone density for each pixel.

The following example showcases a patch projection on the grid to produce a patch with pre-defined location points:

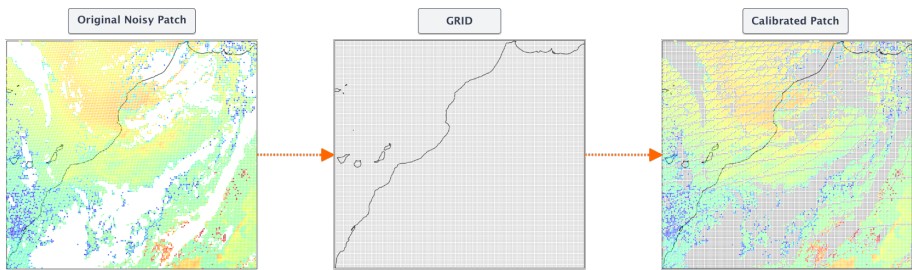

**Figure 2.** Projecting raw scans into an equidistant Grid is important for The CNN Distance Sensitivity

The chosen area of interest is Morocco, Matrices sizes were reduced to match the Moroccan Region, and after getting scans that cover the surface of Morocco, it was still difficult to do any training because the dimensionality of the scans is large, that's why a patches extractor was built to handle the extraction of sub-regions in Morocco and to export pre-processed patches with

15 full coverage over their respective regions.

Exporting full patches is important in the training and testing phases of building the neural network model because having missing data in the training set will cause us to sough unsupervised learning methods to interpolate missing pixels.

By using an Equidistant Grid, the distance between neighboring points is constant and the position of each pixel is known to us.

## 2.4 Adding Noise

Turning raw ground-truth imagery into input-to-output data sets is an interesting problem space, input imagery should simulate data captured by the satellite sensor, satellite noise has an almost smooth continuous surface that simulates visibility issues when collecting measurements (Natural Phenomena like Clouds, Haze, Fog, ..).





To build the data set, a GAN (Generative Adversarial Network) Goodfellow et al. (2014) was trained on the noise distribution to create artificial but similar damaged masks, Examples to follow:

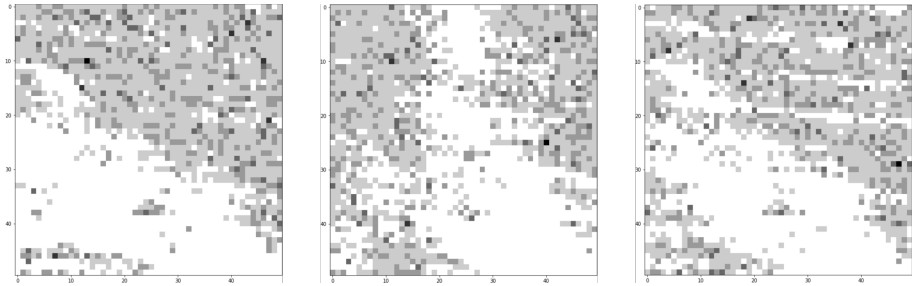

**Figure 3.** Random blocks of the patches were nullified to create the training data set

Although there is no access to a proper supervised learning data set (Natural damaged pixels and their corresponding true measurements), artificial noise was created to turn this problem into a self-supervised learning problem.

The extracted noised patches represent the initial input to the C-SoTa (Classical State-of-the-art) algorithm that will be first interpolated before getting passed to OzoNet.

### 2.5    Classical Algorithms Benchmarking and Results

Next step is to figure out which classical interpolation algorithm to be used to fill in the noised ozone patches.

the fact that the chosen state of the art method will export input data to the OzoNet Model makes the choice of this "base"

algorithm critical to the final OzoNet trained model.

2-D interpolation methods' performance relies heavily on the domain of application, we want to find out which of the classical interpolation algorithms works best to solve our specific problem, we will be using the output of the State-of-the-art algorithm to feed as input to our OzoNet Model, reasonably, we want to first find out which SOTA algorithm works best to then construct a new data set of interpolated Patches to be fed at training time to the OzoNet Model.

Let's call the chosen algorithm C-SOTA (Classical-State-of-the-Art), the following figure demonstrates how a new damaged patch will be interpolated:





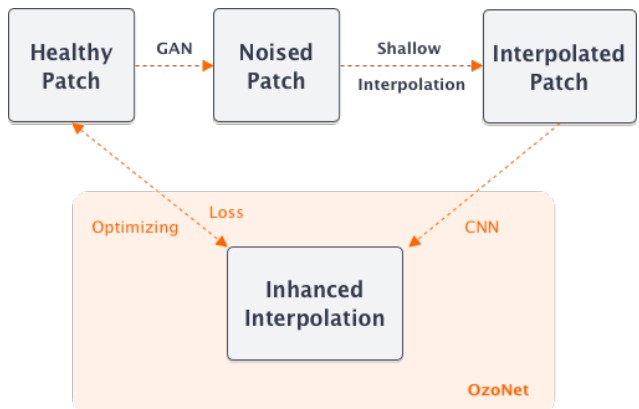

**Figure 4.** Interpolations are produced using both the SoTa Classical Model and the Neural Model

### 2.5.1 State-of-the-Art 2-D Interpolation Algorithms

Three State-of-the-art Industry-wise 2-D Image Processing Algorithms were considered for benchmarking:

– Cubic Spline Interpolation: this method is used primarily for offline image processing, where the quality of the resulting interpolation is the only factor to consider, the resulting image interpolations are usually very satisfying but the underlying complexity of this method makes it slow to process lots of patches or big images.

– Linear Interpolation: This method of interpolation represents a special case of Polynomial Interpolation. It uses linear polynomials to construct new 2-D data points within the range of a discrete set of known data points.

– Nearest Neighbors: This method is used to approximate a point by using the neighboring points in the four 2-D directions. Usually, it results in poor interpolated images, however, this method is utilized in industry for its speed and performance. This method was considered while evaluating speed versus performance, it should be noted that Nearest Neighbors is a bad choice for offline image processing.

### 2.5.2 Scoring

For near-real-time pollution interpolation, the quality of the interpolations and the latency are equally important, having a system that can export ready to be visualized and analyzed pollution patches in near-real time is crucial for the advances in Atmospheric Pollution monitoring at MDEO.

Each of the three Interpolation algorithms was benchmarked based on two factors, accuracy and latency, and scoring consists of simply adding up the two metrics, $A + L$, defined as follows:

– Accuracy: is equal to the Mean Squared Error (MSE) between each interpolated pixel and ground truth original patch pixels, averaged over each patch, then over the 1M patches data set.

– Latency: Average duration taken by the algorithm to interpolate a full patch.





### 2.5.3 Results

Benchmarks were conducted on three different patch dimensions, patches of 30 by 30 pixels, 50 by 50 pixels, and 70 by 70 pixels, as showcased in the following figure:

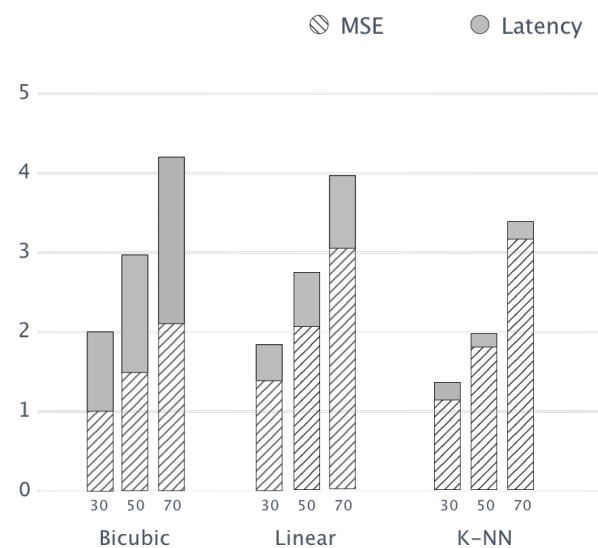

**Figure 5.** Benchmark Results for different State-of-The-Art Classical Interpolation Algorithms

Although the nearest neighbor algorithm has worse results in terms of Mean-Squared-Error, its latency is minimal and makes

5   up for the slightly worse MSE scores.

As a result of that, the nearest neighbor method was chosen to produce input data for OzoNet, C-SOTA will produce interpolations that will be fed to OzoNet in production as well, coupling the two algorithms will serve as MDEO's Ozone interpolator.

## 3   Model Architecture

10   ### 3.1   Problem

The Neural Network Model must interpolate ozone measurements with improved accuracy compared to SoTA 2D Interpolation algorithms, we assume $f$ is the Neural Network Universal Function, $f$ should improve on the following mapping:

$$X^{'} \xrightarrow{f} Y$$

Where $X^{'}$ represents the C-SoTA Interpolated Patches, and $Y$ is the Set of ground-truth patches with healthy pixels.





Patches that are already interpolated using C-SoTA are fed to OzoNet to shift the values of all the pixels in a way that approximates the target pixel measurements.

## 3.2 Architecture

The Architecture is comprised of multiple stacked Convolutions with no Pooling layers, to preserve the output dimensionality

Tishby and Zaslavsky (2015).

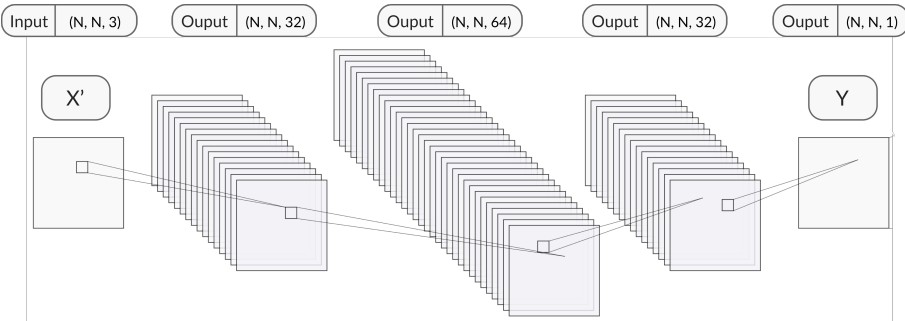

**Figure 6.** The Model Uses Stacked Convolution and Act as a Regressor over the Damaged points

The OzoNet model takes as input the already C-SoTA interpolated patches and outputs enhanced interpolations closer to the ground truth full patches.

OzoNet is comprised of 3-5 Convolution layers (for different patch sizes) with the following specifications:

– Filters: 64, 32, 1.

– Activation Function: The ReLU activation function was used for all layers except the last one (which is Linear).

– Kernel Sizes: sizes 2,3,5,7 were used interchangeably for the various configurations for the 30 by 30, 50 by 50, and 70 by 70 patches.

No-pooling was used on the Architecture, a critical concept that the network was built upon is the fact that we don't want to compress information (opposite to classification tasks), the goal is to rearrange pixel information and preserve patch dimensions

to export the same Image with better interpolated missing/damaged pixels.

## 3.3 Hyper-Parameters

OzoNet has the following hyper-parameters:

– Optimizer: OzoNet uses the RMSProp Optimizer, with a learning rate varying from 0.001 to 0.01.

– Loss function: the same metric to score the classical learning interpolators was used, Mean Squared Error.





- Evaluation Metrics: Because we've set benchmarks using MSE, we've used the Loss to track performance and the state of the model while training.

The Input/Output data matrices are float32 tensors, scaled to values between -1 and 1, and normalized to accelerate convergence time.

## 3.4 Ground Truth Output Reset

A Problem that was encountered during testing is that OzoNet slightly changes the values of the already available pixels to create a smoother surface and to optimize the overall MSE, this happened because the network doesn't detect damaged pixels since it receives full interpolated patches.

What was needed for the network to do is to ignore available surfaces, detect and enhance damaged regions of the images, that's why the outputted images were altered to correct its behavior:

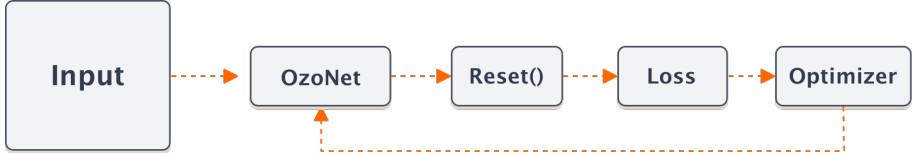

**Figure 7.** By Altering the output of the network, Training is accelerated.

As a way to force the network to detect and improve on the damaged zones, A hook was implemented at the output gate of the last layer to reset ground-truth pixels to their initial values, this resulted in a fast convergence to State-of-the-art scores at the start of the training. It should be noted that training still took time to generalize beyond the training data to output similar scores on the validation set to reach the state of overfitting.

## 3.5 C-SOTA Head Start

The nature of the data gives OzoNet a head start in terms of interpolation accuracy, the kNN interpolated images are already good enough to be considered for production use, and as a direct result, OzoNet was trained from the start to outperform the Classical state-of-the-art approach and produce better interpolations with minimal latency.

After normalizing, interpolating and inputting the patches in batches, little overfitting was observed and the results were interpretable from the start.

## 3.6 Regularization

Multiple learning rates, optimizers, and loss functions were used to optimize the models with slightly worse to having no effect on the results, it was necessary to produce multiple models that are optimized and specific to the different pollution image dimensions, we'll start by outlining the different techniques used for the three corresponding models:

1. OzoNet30





- 3 Convolution Layers.

- 1 Dropout Layer.

- Batch Normalization for all layers except the last one.

2. OzoNet50

- Increased the Convolution layers to 5.

- 2 Dropout Layers.

- Batch Normalization.

3. OzoNet70

- Increased the Convolution Count to 7.

- 3 Dropout Layers.

- Batch Normalization.

- Cyclical Varying Learning Rate Smith (2017).

Enhancing OzoNet's Architecture and changing the hyper-parameters served as a way to fight overfitting and improve the accuracy of the corresponding models.

The resulting models have, in order:

1. OzoNet30: $\approx 500K$ weights.

2. OzoNet50: $\approx 1M$ weights.

3. OzoNet70: $\approx 1.3M$ weights.

## 4   Experiments and Results

To benchmark OzoNet against the classical methods of 2-D interpolation, A hidden test data set was used, comprised of 100,000 hidden patches that represent future scans of the Moroccan region, OzoNet was trained on past patches and the decision to hide a data set comprised of future scans was motivated by simulating production needs (the need for the model that predicts future incoming patches having learned from a fixed data set from the past).

### 4.1   Training

The Data was pre-processed then uploaded to Google Cloud's bucket Storage to have easy access from the GPU servers. The Model Architecture was built using Tensor-Flow Abadi et al. (2016) and trained on two Tesla K80 GPUs with the following specifications:





- 4992 NVIDIA CUDA cores with a dual-GPU design.

- 24 GB of GDDR5 memory.

- 480 GB/s aggregate memory bandwidth.

## 4.2 Benchmarks

Although GPUs were needed in training OzoNet, testing was conducted using CPUs as the weights have already converged and loaded into memory, specifications are close to a normal compute server specifications.

We have two figures that complement each other, the first one represents a detailed overview table in which we can find all of the scores of the 3 classical algorithms and OzoNet, and the second one represents a bar plot visualization of the scores, as follows:

**Table 1.** Benchmarking Results

| Dimensions | 30x30 | | 50x50 | | 70x70 | |
|---|---|---|---|---|---|---|
| Algorithm \| Metric | MSE | Latency | MSE | Latency | MSE | Latency |
| Bicubic | 1.1 | 0.8 | 1.6 | 1.5 | 2.3 | 2.0 |
| Linear | 1.5 | 0.3 | 2.1 | 0.7 | 2.9 | 1.1 |
| Nearest Neighbors | 1.2 | 0.08 | 1.9 | 0.09 | 3.3 | 0.09 |
| **OzoNet** | **0.6** | **0.01** | **0.5** | **0.03** | **1.0** | **0.03** |

The Table above represents averaged scores of 15 trials of benchmarks, As it can be observed, OzoNet outperforms the Bicubic Algorithm, the technique of using classical learning algorithms to accelerate convergence might in some ways be compared to how fine-tuning and transfer learning work.

However, it must be emphasized that OzoNet will be coupled with the nearest neighbor algorithm in production, OzoNet exports worse results when provided with raw damaged ozone patches.

The associated bar chart allows us to quickly have an idea of the performance of OzoNet in comparison to other algorithms without comparing numbers:





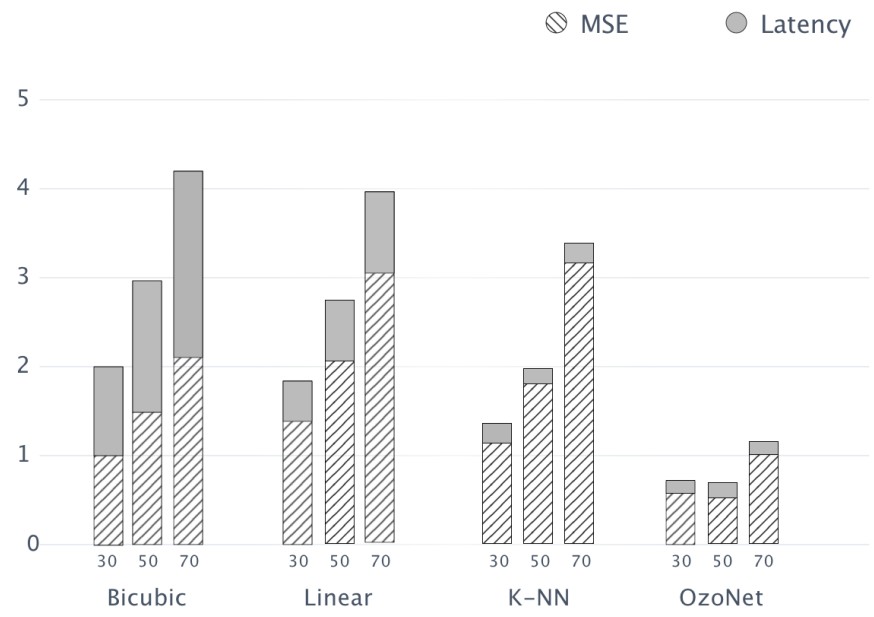

**Figure 8.** Benchmarking After 1 Epoch of Training

OzoNet is fast at exporting the predictions, a noticeable thing is that the loss of its predictions is controlled and its acceleration is minimized, that's an indication of OzoNet's ability to leverage bigger patches, extract vision information, and minimize error in a more efficient way than the classical interpolation methods.

The following figure showcases training progress and the speed at which the model converges to a better MSE accuracy in

5   comparison with the other model:





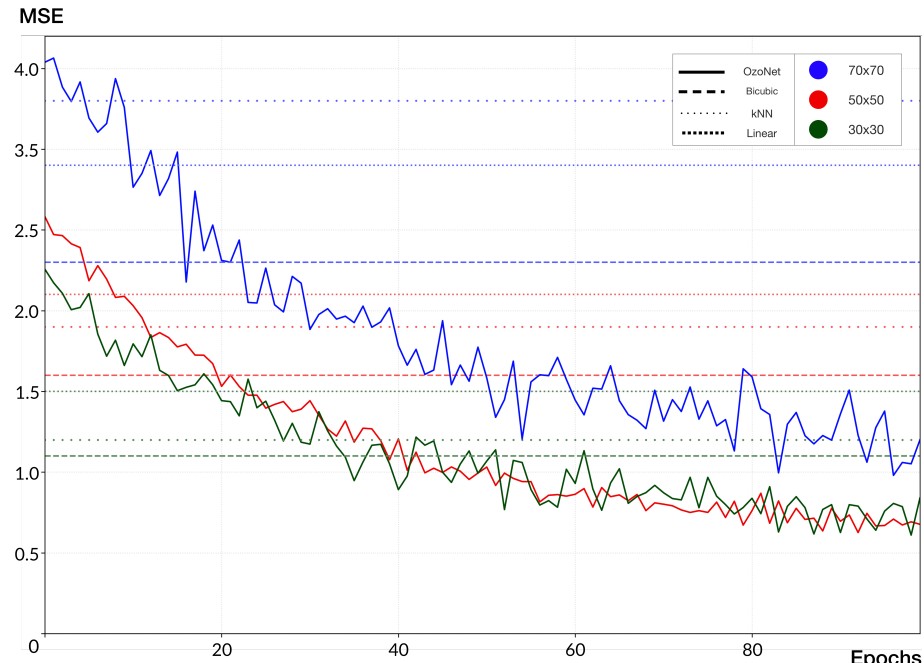

**Figure 9.** OzoNet surpassed SoTa results due to starting with pre-interpolated patches and enhancing them using its convolutions

In training time while optimizing for Mean Square Error, and because OzoNet's Input is the already interpolated images coming from the nearest neighbor algorithm, after training for only two epochs, OzoNet achieves a score better than KNN, and continues to improve to achieve minimal MSE scores that are **50.2%** better than the bicubic slow algorithm (averaged on 30x30, 50x50, and 70x70), **62.3%** better than the Linear Mode, and **58.9%** Better than the KNN Algorithm.

5    **4.3    Results**

We present some examples of OzoNet interpolations in comparison with ground truth and input patches:



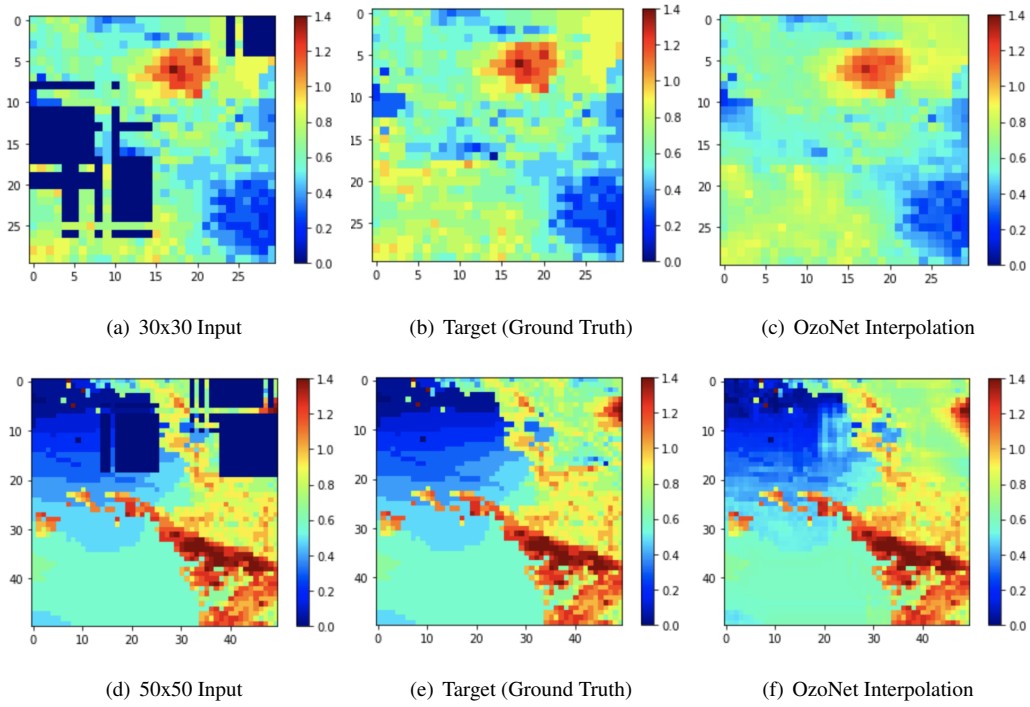

(a) 30x30 Input      (b) Target (Ground Truth)      (c) OzoNet Interpolation

(d) 50x50 Input      (e) Target (Ground Truth)      (f) OzoNet Interpolation

**Figure 10.** Examples of Ozone patches with missing pixel values. The two Patches have different distribution. OzoNet is trained to interpolate the missing values using ground truth patches or targets.

A major improvement over the classical shallow interpolation algorithms is the speed of interpolation over our network, it doesn't have to loop over the missing points to interpolate, it takes one image to produce one interpolation for each patch. and it can make use of GPUs in the case of massive amounts of patches that need to be processed in near-real-time.

## 5 Conclusions

The benchmark results indicate that OzoNet is a faster and more accurate interpolator for remote sensing ozone measurements and for Remote sensing and Atmospheric Imaging In general. Natural Phenomenon like Cloud masking, Fog, and overall visibility problems in remote sensing can be a big burden and obstacle to get a complete view of the atmospheric state in some regions, the proposed solution leveraged a Simple Architecture of Multiple Stacked Convolutions to learn Mappings that are specific to the task at hand.

Deep Learning opens the door to limitless applications in domains such as Climate Science, Agriculture, Air Pollution, Numerical Weather Predictions, and others Deng (2014). Remote sensing data providers also play a central role in the development of the field and in doing practical environmental AI research, by offering huge amounts of data for different purposes that are necessary to train deep neural networks.





We believe that artificial intelligence will be a game-changer for climate change and the overall environmental problems we are facing today. By using high-res imagery, remote sensing numerical data, and satellite data in general, we can better understand atmospheric processes and build systems that contribute in Domains like Energy, Pollution, Agriculture, Oceans, and Climate Science.

5   *Competing interests.*   No competing interests are present.

*Acknowledgements.*   The authors are thankful to the Ministry of Higher Education and Scientific Research, and the National Centre for Scientific and Technical Research (CNRST) for funding this study, under the project: PPR/2015/7.





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
