# Peer review of "OzoNet: Atmospheric Ozone Interpolation with Deep Convolutional Neural Networks"

_Geoscientific Instrumentation, Methods and Data Systems, 2018_

## Referee Comment (RC1) · Anonymous Referee #1 · 4 Jul 2019

The paper discusses the method of interpolation of ozone field with data gaps using convolutional neural networks. Comparison with some traditional interpolation methods is performed. My comments are below.

1) Unfortunately, the paper is written in a very unclear way with poor English. This disturbs strongly the paper reading and understanding. The help of native or advanced English speakers is mandatory. In addition, the style should correspond the style of scientific paper (acronyms and abbreviations should be explained at first appearance; data should be described in the text, while technical details should be shortened and/or collected in tables and appendices (not in itemized list as in the present version e.g. in

[Figure]

page 2); references should be according to Copernicus publication style etc.).

2) The objective for developing the interpolation method of the ozone field should be clearly stated. From the general point of view, filling data gaps by any kind of interpolation might introduce errors. Therefore, necessity of interpolation should be explained. Also some extreme cases should be illustrated, for example, when significant part of a plume is missing in the data. A discussion on data coverage, which is required for accurate interpolation, is also needed.

3) There are advanced interpolation methods that use information about the structure of data field (kriging). It would be advantageous to compare the interpolation results using CNN and kriging.

4) The description of OzoNet system is missing. A more detailed description is needed, with explanation what is contained in the Supplement.

5) I recommend naming cubic, linear and nearest-neighbors interpolation as "traditional methods".

6) Since the OzoNet uses nearest-neighbors interpolated field as an input, the total OzoNet latency is actually the sum of two latencies: from "nearest-neighbors" and from the neural network interpolation.

---

## Editor Comment (EC1) · Walter Schmidt (Editor) · 9 Aug 2019

Dear authors,

All efforts to get additional reviewers for your manuscript failed and I am deeply sorry for the extremely long waiting time, filled with endless trials to get the formal review process first started and then to a conclusive end. I had several e-mail discussions with colleagues working in this field, among them also the former chief editor of our journal. The responses were similar to what the only volunteering reviewer #1 wrote: Though the topic is interesting and fits into the subject area of our journal, the manuscript should not be published in its current form as a reviewed paper. By the official end

of the discussion phase on July 30 there were no additional comments from the community which could have given an opinion deviating from the reviewer #1's evaluation. Therefore I am inclined to follow the suggestions and reject the manuscript in its current form. The main problem is the missing discussion of other used methods to put your approach into context and give the reader an idea of its possible usefulness for similar own applications. Also the level of detail used in your application is not such that it would allow the development of a similar approach elsewhere. These two aspects were the main reasons why this specific journal was originally started.

As I wrote in my original short evaluation both topic and approach are very interesting and could benefit a wide reader community. If you see a possibility to implement the most important suggestions of the reviewer's comments and make the manuscript more applicable to possible readers I would be happy to have a look at a new submission as possible handling associate editor.

Best regards, Walter Schmidt, associate editor